# The Expression of Epac2 and GluA3 in an Alzheimer’s Disease Experimental Model and Postmortem Patient Samples

**DOI:** 10.3390/biomedicines11082096

**Published:** 2023-07-25

**Authors:** Tong Zhang, Nshunge Musheshe, Christina H. J. T. M. van der Veen, Helmut W. Kessels, Amalia Dolga, Peter De Deyn, Ulrich Eisel, Martina Schmidt

**Affiliations:** 1Department of Molecular Pharmacology, University of Groningen, 9713 AV Groningen, The Netherlands; tong.zhang@rug.nl (T.Z.); nshunge@yahoo.co.uk (N.M.); c.h.t.j.mol-van.der.veen@rug.nl (C.H.J.T.M.v.d.V.); a.m.dolga@rug.nl (A.D.); 2Department of Molecular Neurobiology, Groningen Institute for Evolutionary Life Sciences, University of Groningen, 9747 AG Groningen, The Netherlands; u.l.m.eisel@rug.nl; 3Swammerdam Institute for Life Sciences, University of Amsterdam, 1098 XH Amsterdam, The Netherlands; h.w.h.g.kessels@uva.nl; 4Institute for Asthma and COPD, GRIAC, University Medical Center Groningen, University of Groningen, 9713 GZ Groningen, The Netherlands; 5Department of Neurology and Alzheimer Research Center, University of Groningen, University Medical Center Groningen, 9713 GZ Groningen, The Netherlands; p.p.de.deyn@umcg.nl; 6Laboratory of Neurochemistry and Behavior, Experimental Neurobiology Unit, University of Antwerp, 2610 Wilrijk, Belgium

**Keywords:** Alzheimer’s disease, Epac2, AMPARs, GluA3, AKAP5, PSD95, PI3K/Akt, ERK1/2

## Abstract

Alzheimer’s disease (AD) is one of the most prevalent neurodegenerative diseases, characterized by amyloid beta (Aβ) and hyperphosphorylated tau accumulation in the brain. Recent studies indicated that memory retrieval, rather than memory formation, was impaired in the early stage of AD. Our previous study reported that pharmacological activation of hippocampal Epac2 promoted memory retrieval in *C57BL/6J mice*. A recent study suggested that pharmacological inhibition of Epac2 prevented synaptic potentiation mediated by GluA3-containing AMPARs. In this study, we aimed to investigate proteins associated with Epac2-mediated memory in hippocampal postmortem samples of AD patients and healthy controls compared with the experimental AD model *J20* and *wild-type mice*. Epac2 and phospho-Akt were downregulated in AD patients and *J20 mice*, while Epac1 and phospho-ERK1/2 were not altered. GluA3 was reduced in *J20 mice* and tended to decrease in AD patients. PSD95 tended to decrease in AD patients and *J20*. Interestingly, AKAP5 was increased in AD patients but not in *J20 mice*, implicating its role in tau phosphorylation. Our study points to the downregulation of hippocampal expression of proteins associated with Epac2 in AD.

## 1. Introduction

Alzheimer’s disease (AD) is a progressive neurodegenerative disease that represents the most prevalent form of dementia. It is currently ranked as the fifth leading cause of mortality, affecting approximately 45 million individuals globally [1]. According to the yearbook of Dementia in Europe, the current prevalence of AD is 1.57% of the total European population, and this number is estimated to double to 3% by the year 2050 [2]. The most significant risk factor for sporadic AD is age, which accounts for most AD cases. Age shows a strong correlation with AD prevalence, with a doubling percentage observed every 5 years among elderly populations [2].

AD is clinically characterized by amyloid beta (Aβ) aggregation, hyperphosphorylated tau, brain atrophy, and memory impairment [3]. Notably, Aβ appears to be the earliest and upstream pathophysiology in AD, up to 30 years prior to the onset of cognitive symptoms [4]. Therefore, Aβ was hypothesized to play an important role in the pathogenesis of AD [5]. Aβ aggregation is initiated in monomers, which stepwise form into dimers, oligomers, fibrils, and plaques, with only a small pool of bioactive Aβ oligomers showing neurotoxicity [5]. Neurotoxic Aβ species are proposed to activate glia cells, facilitate tau hyperphosphorylation, trigger synaptic dysfunction and loss, and ultimately lead to AD [6]. In order to investigate how Aβ contributes to AD pathology, several AD transgenic models, such as *APP/PS1* and *J20*, were developed overexpressing Aβ precursor protein (APP) and displaying early memory deficits [7].

Compared with late-stage AD, early-stage AD exhibits less irreversible neurodegeneration and fewer neuronal lesions, making early intervention easier and potentially improving treatment outcomes. It has been reported that early cognitive training facilitates retrieval of long-term memory in *APP/PS1 mice* at the age of 7 months [8], which is considered an early AD model due to its memory decline in the absence of (Aβ plaques and) neurodegeneration [9]. Increasing evidence further demonstrates that retrieval, rather than formation, of episodic memory is impaired in early-stage AD [9,10]. To address this, Roy et al. optogenetically activated memory retrieval in engram neurons from dentate gyrus, resulting in a significant memory improvement in 7-month-old *APP/PS1 mice* [9]. Therefore, targeting impaired memory retrieval could be a promising strategy to alleviate memory deficits in early-stage AD. 

Modulation of cyclic adenosine monophosphate (cAMP)/exchange factor directly activated by the cAMP 2 (Epac2) pathway could be a strategy to enhance synaptic plasticity and thereby improve memory deficits in AD. As a main cAMP effector, Epac2 has been found to mediate presynaptic neurotransmission in hippocampal neurons [11] and postsynaptic plasticity in Purkinje cells (PCs) [12]. Our lab has previously revealed that short-term (20 min) activation of Epac2 in the hippocampal CA1 region promoted memory retrieval in *C57BL/6J mice* [13]. Similarly, another study reported that activation of protein kinase A (PKA) and Epac during memory retrieval in the dorsal hippocampus rescued memory deficit in a KO mouse model of dopamine hydroxylase [14]. A study using Epac2 KO mice further confirmed that Epac2 was involved in memory retrieval [15]. These studies collectively suggest that Epac2 could be a promising target to rescue impaired memory retrieval in the early stage of AD.

The cAMP pathway is involved in mediating the function and trafficking of α-amino-3-hydroxy-5-methyl-4-isoxazolepropionic acid receptors (AMPARs), which play key roles in synaptic plasticity and memory [16]. Studies have demonstrated that the cAMP effector PKA phosphorylates GluA1 subunits of AMPARs, which directly promote synaptic trafficking and anchoring of GluA1-containing AMPARs in postsynaptic density 95 (PSD95) [16,17,18]. A-kinase anchoring protein 5 (AKAP5, AKAP79 in *humans*, AKAP150 in *mice*) is also involved in anchoring PKA to GluA1 subunits [19]. In addition to GluA1-mediated plasticity, cAMP is found to trigger novel synaptic plasticity involving GluA3-containing AMPARs [20]. Gutierrez et al. reported that GluA3-containing AMPARs mediated basal synaptic transmission and long-term potentiation (LTP) in PCs, indicating an alternative type of AMPAR plasticity [12]. This GluA3-dependent LTP was induced by the adenylyl cyclase activator forskolin and blocked by the Epac2 inhibitor ESI-05, but not by protein kinase A (PKA) inhibitors H89 and KT5720, suggesting that cAMP/Epac2 was involved in GluA3 plasticity [12]. Similarly, cAMP-driven GluA3 plasticity was also observed in CA1 hippocampal neurons, which was mediated by Ras after activation of β-adrenergic receptors [21]. These studies, performed by our research team, indicate a mechanism for cAMP signaling in GluA3-dependent synaptic plasticity.

In AD, the extracellular accumulation of neurotoxic Aβ oligomers exhibits a significant impact on glutamatergic neurotransmission, leading to dysfunctional synaptic transmission and plasticity [3]. Reinders et al. reported that GluA3-deficient neurons exhibited resistance to synaptic depression and spine loss induced by Aβ, while GluA3-expressed neurons showed impaired LTP mediated by Aβ. Furthermore, in contrast to significant memory impairment observed in *APP/PS1 mice*, *GluA3 KO mice* did not show any such deficits in memory [22]. These results suggest that synaptic depression in AD is initiated by the Aβ-mediated removal of GluA3-containing AMPARs.

Epac2 signalosome has been implicated in synaptic plasticity and memory [23,24]. As an anchoring protein in Epac2 signalosome, AKAP5 was found to coordinate Epac2 and PKA in primary neurons, which respectively upregulated and downregulated phospho-Akt (Ser473) (p-Akt) [25], a downstream pathway of Epac2 signaling involving in synaptic plasticity and memory [26,27,28,29]. Apart from p-Akt, phospho-ERK1/2 (Thr202/Tyr204) (p-ERK) is another pathway regulated by Epac2, which is implicated in synaptic plasticity [30]. As a scaffolding protein of AMPARs, PSD95 has been reported to interact with AKAP5 and facilitate trafficking of AMPARs [31]. Therefore, we propose an Epac2-mediated postsynaptic complex that is coordinated by AKAP5 and PSD95 (Figure 1).

Several studies have explored the mRNA and protein expression levels of Epac2 and GluA3 in various cell and animal models [38,39]. However, and most important, there is still a lack of translational research conducted in postmortem samples of AD patients, particularly in the hippocampus. Previous studies reported reduced protein levels of hippocampal GluA1 and PSD95 in AD and patients with mild cognitive impairment (MCI), respectively [40]. Interestingly, elevated levels of GluA3 protein were observed in the cerebrospinal fluid (CSF) of AD patients, which was correlated with cognitive impairment and tau pathology [40]. In addition, mRNA levels of Epac1 and Epac2 were upregulated and downregulated, respectively, in frontal cortex postmortem samples of AD patients [41], while Epac anchoring protein AKAP5 showed upregulation in whole brain homogenates of AD patients [37]. Given the need for translational studies, we investigated here hippocampal proteins associated with Epac2-mediated memory in AD patients and *J20 AD mice*.

## 2. Materials and Methods

### 2.1. Human Samples

Human hippocampal brain specimens collected from healthy controls and AD patients were provided by the Biobank of the Institute Born-Bunge (University of Antwerp, Belgium). The samples were preserved at −80 °C for Western blot. All participants granted informed consent for both the behavioral assessments and the postmortem tissue collection. The characteristics of healthy controls and AD patients are shown in Appendix A. The use of human samples was approved by the Medical Ethical Committee of the Middelheim General Hospital (Antwerp, Belgium) under approval numbers 2805 and 2806.

### 2.2. J20 Samples

The *J20* mouse is a *C57Bl/6 mouse* line that overexpresses human APP with Indiana and Swedish mutations and was used as the experimental model of AD [42]. This mouse line originated from Mutant Mouse Resource and Research Center (MMRRC stock no. 034836-JAX; former JAX stock no. 006293, Maine, ME, USA) and was bred with water and food ad libitum under 12-h light/dark cycles in our facility. Hemizygous *J20* was genotyped using polymerase chain reaction (PCR). The hippocampus was isolated from male mice at the age of 8–12 months. All animal procedures were approved by the animal ethics committee of the University of Groningen (DEC6851A).

### 2.3. Western Blot

Human and mouse hippocampal samples were homogenized in a modified RIPA buffer (0.05 M Tris-Base (1185-53-1, Sigma-Aldrich), 0,25 M D-Manitol (240184, Sigma-Aldrich, Saint Louis, MI, USA), 1 mM EDTA (E6758, Sigma-Aldrich, Saint Louis, MI, USA), 1 mM EGTA (E4378, Sigma-Aldrich, Saint Louis, MI, USA), 0.25% Na-deoxycholate: 0.25%, pH = 7.8) supplemented with 1 mM DTT (3483-12-3, Sigma-Aldrich, Saint Louis, MI, USA), 0.2% Triton x-100 (X100, Sigma-Aldrich, Saint Louis, MI, USA), and protease and phosphatase inhibitors. The protein samples were collected after centrifugation at 12,000 rcf for 15 min at 4 °C and determined by BCA assays (Pierce, Thermo Scientific, Waltham, MA, USA). A total of 20–40 µg of lysates was loaded onto 10% SDS-PAGE gels and transferred to PVDF or nitrocellulose membranes. The membranes were subsequently blocked with 2.5~5% low-fat milk or BSA in Tris-buffered saline with 0.1% Tween-20 (P1379, Sigma-Aldrich, Saint Louis, MI, USA) for 60 min at room temperature (RT), followed by incubation with primary antibodies at 4 °C overnight. Primary antibodies included mouse anti-GluA3 (1:1000 dilution, MAB5416, Sigma-Aldrich, Saint Louis, MI, USA), rabbit anti-PSD95 (1:1000 dilution, #2507, Cell Signalling, Danvers, MA, USA), mouse anti-Epac2 (1:1000 dilution, 5B1, Cell Signalling, Danvers, MA, USA), mouse anti-Epac1 (1:1000 dilution, 5D3, Cell Signalling, Danvers, MA, USA), rabbit anti-phospho-Akt (Ser473) (1:1000 dilution, #9271, Cell Signalling, Danvers, MA, USA), rabbit anti-pan-Akt (1:1000 dilution, #4691, Cell Signalling, Danvers, MA, USA), rabbit anti-phospho-ERK1/2 (1:1000 dilution, #9101, Cell Signalling, Danvers, MA, USA), rabbit anti-ERK1/2 (1:1000 dilution, #9102, Cell Signalling, Danvers, MA, USA), mouse anti-AKAP79 (1:500 dilution, sc-17772, Santa Cruz, Santa Cruz, CA, USA,), mouse anti-AKAP150 (1:1000 dilution, sc-377055, Santa Cruz, Santa Cruz, CA, USA), and mouse anti-β-actin (1:3000 dilution, sc-47778, Santa Cruz, Santa Cruz, CA, USA). On day 2, the membranes were incubated with secondary antibodies (#A9044, rabbit anti-mouse, 1:3000, Sigma-Aldrich, Saint Louis, MI, USA; GTX2131, goat anti-rabbit, 1:3000, GeneTex, Taiwan, China) for 1 h at RT and were developed in ECL reagent (203-21141, PerkinElmer, Shelton, CT, USA).

### 2.4. Statistical Analysis

Unpaired Student’s T test was used to evaluate statistical significance. Linear regression was used to investigate trends of protein expressions with age. All statistical analyses were two-tailed. The differences were considered statistically significant when *p* ≤ 0.05. Statistical significance of *p* values in results was shown by asterisks as follows: **p* ≤ 0.05; ** *p* ≤ 0.01; *** *p* ≤ 0.001, or not significant (ns). Data were expressed as mean ± standard error of the mean (SEM). All statistical analyses were conducted in GraphPad Prism 7.0 (GraphPad Software, San Diego, CA, USA).

## 3. Results

### 3.1. Epac2, but Not Epac1, Was Downregulated in AD Patients and AD Model Mice

To investigate whether protein levels of Epac1 and Epac2 were altered in AD, we used Western blots on postmortem hippocampal tissue from AD patients and health controls. Epac1 protein levels were not significantly altered between AD patients and controls (Figure 2B). In contrast, hippocampal Epac2 was significantly downregulated in postmortem samples of AD patients compared with controls (Figure 2A). Interestingly, in contrast to healthy controls, protein expression of Epac2 in postmortem samples of AD patients decreased, largely independent of age (Figure 2C). On the other hand, Epac1 gradually decreased with age in both AD patients and healthy controls (Figure 2D). To assess whether Epac expression levels were also affected in mouse models for familial AD, hippocampi from 8–12-month-old *J20 mice* were isolated. Similar to AD patients, *J20 mice* showed a decrease in Epac2 levels, but not Epac1 levels (Figure 2E,F). These results indicate that AD is associated with a decreased expression of Epac2 in the hippocampus.

### 3.2. AKAP5 Was Upregulated in Postmortem Samples of AD Patients

Next, we evaluated expression levels of the Epac2 anchoring protein AKAP5. In postmortem samples of AD patients, AKAP5 protein levels were significantly upregulated compared with controls (Figure 3A). Interestingly, AKAP5 protein expression gradually decreased with age in postmortem samples of AD patients, similar to the age-related expression pattern of Epac2 (Figure 3B). This comparable decrease in age-related expression levels potentially points to the Epac anchoring properties of AKAP5 as reported earlier [25]. However, AKAP5 protein expression was not significantly altered in the experimental AD model *J20 mice* (Figure 3C).

### 3.3. Protein Levels of GluA3 and PSD95 in AD Patients and AD Models

Studies in mouse models have shown that AMPAR subunit GluA3 and the AMPAR anchoring protein PSD95 determine the susceptibility of synapses for Aβ [22,43]. Hippocampal protein expression of GluA3 tended to be downregulated in postmortem samples of AD patients, but this decrease did not reach statistical significance (Figure 4A). PSD95 protein expression also tended to decrease in postmortem samples of AD patients without reaching statistical significance, with a rather variable expression in healthy controls (Figure 4B). The protein levels of GluA3 and PSD95 decreased with age in both postmortem samples of AD patients (Figure 4C), potentially pointing to the complex formation properties of PSD95 as reported earlier [31]. In the experimental AD model *J20 mice*, GluA3 protein expression significantly decreased in comparison to WT mice (Figure 4D), and protein expression of PSD95 tended to de downregulated, with a rather variable expression in the *J20* group (Figure 4D). In conclusion, GluA3 and PSD95 protein levels were similarly affected in the hippocampi of AD patients and AD-model mice.

### 3.4. p-Akt (Ser473), but Not p-ERK1/2 (Thr202/Tyr204), Were Downregulated in AD Models

Two downstream pathways of Epac2, p-Akt and p-ERK, were investigated. In postmortem samples of AD patients, hippocampal p-Akt significantly decreased (Figure 5A), whereas p-ERK was not significantly altered (Figure 5B). p-Akt and p-ERK decreased with age in postmortem samples of AD patients and healthy controls (Figure 5C,D). Similarly, in the experimental AD model *J20 mice*, p-Akt was significantly reduced (Figure 5E), while p-ERK was not significantly altered (Figure 5F).

## 4. Discussion

Recent studies [44] provided evidence that memory retrieval, not memory formation, is impaired in the early stage of AD [9]. Our group has identified that pharmacological activation of Epac2 specifically improved memory retrieval [13]. Until now, only a few reports have focused on changes in Epac2 expression in models of AD [32]. Therefore, in this study, we investigated the hippocampal expression of a protein associated with Epac2-mediated memory in postmortem samples of AD patients and the experimental AD model *J20* (Figure 1).

Studies in the past decade have used the general Epac activator 8-pCPT-2′-O-Me-cAMP (8-pCPT) to modulate the activation of Epac1 and Epac2, the latter of which has been associated with improved memory retrieval [13,14]. A previous study reported a decreased mRNA level of Epac2 and an increased mRNA level of Epac1 in frontal cortex postmortem samples of AD patients [41]. Therefore, we investigated the protein expression of Epac1 and Epac2 in hippocampal postmortem samples of AD patients and the experimental AD model *J20 mice*. In human hippocampal postmortem samples, we observed a significant downregulation of Epac2 in AD patients and no significant difference of Epac1. In *J20 mice*, we also observed downregulation of Epac2, but not of Epac1, indicating that the loss of Epac2 is characteristic of early AD and unlikely to be a consequence of neurodegeneration. Although we observed no change in Epac1 levels in hippocampal postmortem samples of AD patients, an upregulation of Epac1 mRNA was previously reported in frontal cortex postmortem samples of AD patients [41]. Potentially Epac1 shows distinct alterations in different brain areas (frontal cortex [41] versus hippocampus), but mRNA and protein levels are not similarly affected. Future studies should focus on these specific distinctions between Epac1 and Epac2. Interestingly, the downregulation of Epac2 protein seems to associate primarily with AD rather than age, whereas Epac1 protein seems to decline in samples from AD patients and healthy controls “simply” depending on age. Collectively, our data suggest that decreased protein expression of Epac2, rather than of Epac1, is linked to AD. Our findings open the possibility that impaired memory retrieval in AD may be associated with a decline in Epac2 protein levels.

AKAP5 is an anchoring protein for Epac2 and PKA and is highly expressed in brain regions associated with memory and learning [25,45]. Previous studies reported that AKAP5 via its coupling with PKA mediated AMPAR phosphorylation and surface targeting during homeostatic plasticity [18], while Aβ oligomers disrupted AKAP5 coordinated plasticity [19]. Interestingly, in contrast to the decreased protein expression of Epac2, our data showed that AKAP5 was significantly upregulated in hippocampal postmortem samples of AD patients, which was consistent with a previous study performed in hippocampal postmortem samples from AD patients [37]. It was hypothesized that the upregulation of AKAP5 protein in AD patient samples may facilitate hyperphosphorylation of tau by PKA [27,33]. Intriguingly, our experimental AD model *J20 mice* showed no upregulation of AKAP5 protein, further suggesting that AKAP5 is (primarily) linked to tau pathology in AD. As we observed similar age-related expression patterns of AKAP5 and Epac2 in hippocampal postmortem samples of AD patients (Figure 3C), it is tempting to speculate that the upregulation of AKAP5 might represent a compensatory mechanism due to the loss of Epac2. However, studies into underlying regulatory mechanisms of the interplay of AKAP5 and Epac2—potentially acting in concert with PKA—are far beyond the scope of our current manuscript. Further investigations are still needed to elucidate the expression of the Epac2/AKAP5 complex at the synapse.

Recent studies reported that the elevation of cAMP by the adenylyl cyclase activator forskolin triggered synaptic plasticity mediated by GluA3-containing AMPARs [21,22], which were selectively targeted by Aβ [21,22]. Additional studies found that the Epac2 inhibitor ESI-05 blocked the activation of GluA3-containing AMPARs [12,21]. Here we showed that hippocampal protein expression of GluA3 tended to be downregulated in postmortem samples of AD patients, whereas GluA3 protein expression significantly decreased in the experimental AD model *J20 mice*. Our data may indeed implicate that hippocampal GluA3 confers sensitivity to Aβ. As the protein expression of GluA3 tends to decrease with age in both postmortem samples of AD patients and healthy controls (Figure 4e), our studies are potentially challenged by the heterogeneity in the age of the hippocampal AD samples being analyzed (for details see Appendix A) and the highly dynamic nature of synaptic AMPAR trafficking [46]. Interestingly, it has been reported that protein expression of GluA3 decreased in hippocampal synaptosomes of AD patients [47] and increased in CSF of AD patients [40]. It is tempting to speculate that AD triggers the dissociation of GluA3 from synapses, and/or even causes programmed neuronal death (see p-Akt below), leading to its subsequent release into CSF.

PSD95 serves as a key anchoring protein to stabilize AMPARs at the postsynaptic membrane, thereby facilitating the maintenance of synaptic plasticity [16]. It has been reported that an increased expression of PSD95 occurs in the hippocampus or amygdala after the retrieval of recent or remote fear memories, respectively [28]. A previous study reported on a reduction of approximately one-third of PSD95 in MCI patients [48]. In addition, PSD-95 helps to protect synapses from Aβ-mediated synaptic depression [43]. Our data show a tendency toward a decrease in PSD95 protein levels in hippocampal postmortem samples of AD patients, possibly indicating the loss of PSD95 along with AD progression. Intriguingly, as the protein expression of GluA3 and PSD95 tends to decrease with age in both postmortem samples of AD patients, the data further point to the complex formation properties of AKAP5 and PSD95 as reported earlier [31].

The p-Akt (PI3K/Akt) pathway is a key pathway that is involved in neuroprotection and neuronal survival in AD [49,50], and earlier reports have linked Epac to p-Akt [32,33]. In addition, the p-Akt pathway has been implicated in contextual memory formation and retrieval in rodents [26,27,28], next to the regulation of neuronal plasticity and LTP in hippocampal neurons [29]. Furthermore, some studies suggest that PI3K directly interacts with AMPARs and, thereby, regulates AMPARs insertion into postsynaptic membranes during LTP [29,51]. It is worth noting that the Akt phosphorylation during memory retrieval is transient (approximately 2 h) [26]. Our data show a significant decrease in hippocampal p-Akt pathway in AD patients (here correlated with age, Figure 5E) and *J20 mice* (Figure 5A). As p-Akt is linked to neuronal survival and memory, our data may implicate that the loss of p-Akt gives rise to neuronal death and impaired memory in AD.

Studies have demonstrated that Epac can activate ERK1/2 in hippocampal neurons [34], and application of the Epac activator 8-pCPT leads to enhanced LTP maintenance through a transient increase (for less than 10 min) in p-ERK1/2 levels in the CA1 region of hippocampal slices [30]. A recent computational study suggested that the Epac-induced p-ERK1/2 was the main source of transient p-ERK1/2, occurring between 200 and 400 s [52], which may contribute to Epac2-mediated memory retrieval. Indeed, it was reported that inhibition of ERK1/2 phosphorylation impaired memory retrieval [53]. A recent study further revealed that transient ERK phosphorylation during memory retrieval promoted the transition from memory reconsolidation to memory extinction [54]. Interestingly, inhibition of hippocampal p-Akt led to transient downregulation of p-ERK1/2 and impaired memory retrieval, indicating an interplay between p-Akt and p-ERK in memory retrieval [26]. As we analyzed ERK phosphorylation in postmortem samples of AD patients and *J20 mice*, we provided insight into the “steady-state” activation of ERK. In addition, decreased p-ERK1/2 in AD patients and healthy controls seems to correlate with age (Figure 5E). In *J20 mice* samples, transient phosphorylation was hard to detect (Figure 4D), as we did not collect the samples immediately after memory retrieval measurements due to technical restrictions of our animal protocol.

Technical restrictions hampered the measurements of transient phosphorylation of Akt and ERK1/2 in our current AD models. Certainly, it would be of huge interest to further confirm our findings in follow-up studies. For example, as AKAP5 has been identified in synaptosomes and postsynaptic densities [55], it is intriguing to further elucidate the expression of postsynaptic Epac2/AKAP5 complexes in synaptosomes. Such strategies may help to unravel the existence of different AKAP5 pools and may shed light on the question of whether the upregulation of AKAP5 as observed in our current study may represent primarily its cytosolic pool. In addition, mouse lines that express both tau and Aβ pathologies, such as *3xTg AD mice*, represent a powerful tool to study whether the postsynaptic AKAP5 expression is affected by Aβ and/or tau, given that Aβ deposition (3–4 months) chronologically precedes the tau tangles (7 months) [56]. Follow-up studies will certainly shed more light on the composition of the Epac2 signalosome.

Our current findings point to a potential role of the Epac2 pathway in AD, thereby opening the possibility that pharmacological activation of Epac2 could be a new approach to alleviating memory deficits in AD. However, in previous studies, the use of the Epac1 and Epac2 activator 8-pCPT has been hindered by potential off-target effects and rather low specificity for Epac2 compared with Epac1 [41,54]. To overcome these limitations, highly specific Epac2 activators, including S220 and S223, have been recently developed, showing promise for improving the efficacy of Epac2 activation [57].

## 5. Conclusions

In summary, our study revealed the downregulation of Epac2 and p-Akt, next to an upregulation of AKAP5, in hippocampal postmortem samples of AD patients. Consistently, we also observed the downregulation of Epac2, p-Akt, and GluA3 in the experimental AD model *J20 mice*. Our observation of an upregulation of AKAP5 in AD patients, but not *J20 mice*, may implicate its role in tau pathology in AD. We argue that these findings contribute to our understanding of the underlying molecular mechanisms associated with the impairment of memory retrieval in AD, providing potential targets for future therapeutic interventions.

## Figures and Tables

**Figure 1 biomedicines-11-02096-f001:**
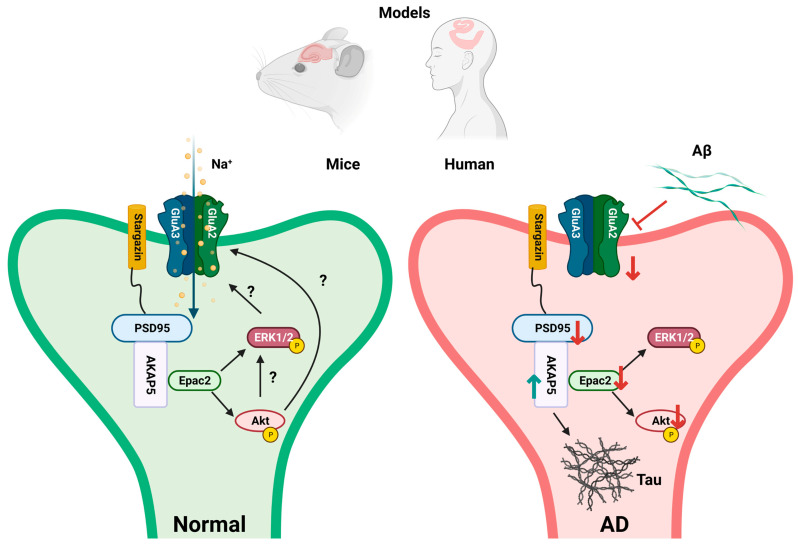
Schematic illustration of postsynaptic Epac2 and GluA3 signaling complexes in the hippocampus. Shown are healthy controls (Normal) and alterations under disease conditions (AD). In postsynaptic areas, Epac2 and AMPARs are anchored by AKAP5 and stargazin/PSD95 [25,31]. Complex formation of AKAP5 and PSD95 [31] potentially facilitates complex formation with Epac2. Epac2 increases phospho-Akt and/or phospho-ERK and, thereby, contributes to synaptic plasticity and memory [30,32,33,34]. In postmortem samples of AD patients, the protein expression of Epac2 and phospho-Akt are downregulated, and GluA3 and PSD95 tend to be downregulated—these processes bear the potential to impair synaptic plasticity. On the other hand, AKAP5, reported to mediate Tau phosphorylation by PKA, is upregulated [35,36,37]. In the experimental AD model *J20* mice, Epac2, GluA3, and phospho-Akt were downregulated. Epac2: exchange factor directly activated by cAMP 2; AMPARs: α-amino-3-hydroxy-5-methyl-4-isoxazolepropionic acid receptors; PSD95: postsynaptic density 95; AKAP5: A kinase anchoring protein 5 (AKAP79/150); Akt: Ak strain transforming; ERK1/2: extracellular signal-regulated kinase 1/2. Model created with BioRender.com.

**Figure 2 biomedicines-11-02096-f002:**
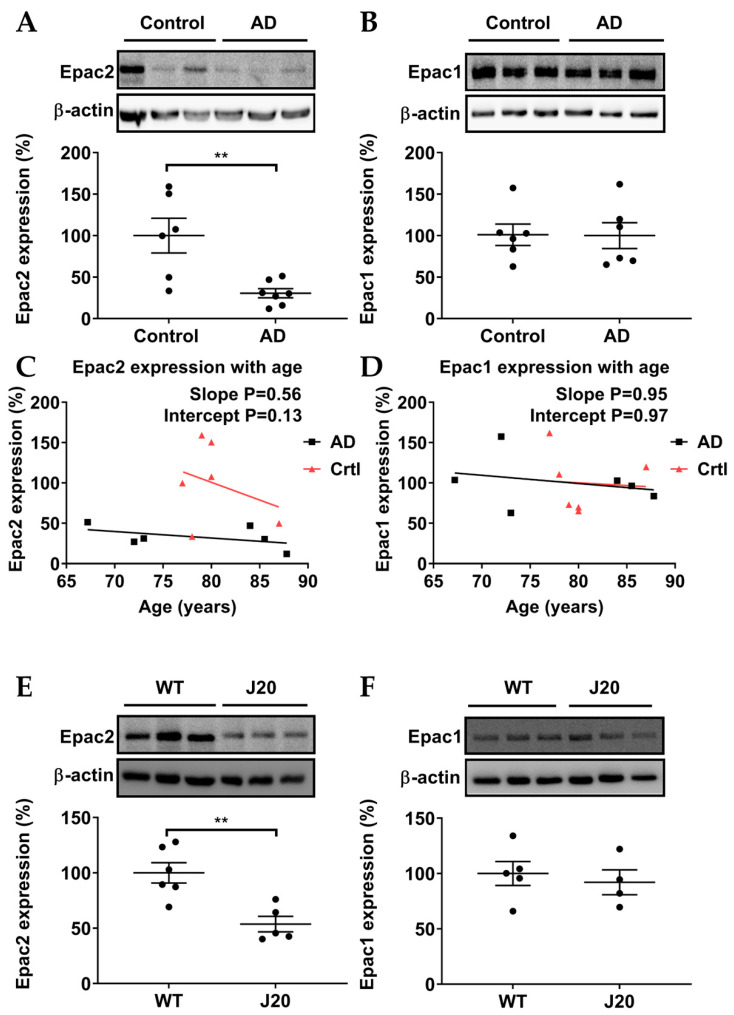
Relative protein expression levels of Epac1 and Epac2. (**A**,**B**) In postmortem samples of AD patients, the protein expression of Epac2 was significantly downregulated while Epac1 was not significantly altered. (**C**) Protein expression of Epac2 significantly decreased in postmortem samples of AD patients largely independent of age, whereas it gradually decreased with age in healthy controls (Crtl). (**D**) Protein expression of Epac1 gradually decreased with age in all postmortem samples. (**E**,**F**) In experimental AD model *J20 mice*, the protein expression level of Epac2 was significantly downregulated while Epac1 was not significantly altered. Quantification and representative blots are shown. Healthy controls n = 6; AD patients n = 6–7; *WT mice* = 5–6; *J20 mice* = 4–5. Data are expressed as mean ± SEM. Unpaired student’s T test and linear regression were used to study protein expressions and the trends of protein expressions with age, respectively (** *p* ≤ 0.01).

**Figure 3 biomedicines-11-02096-f003:**
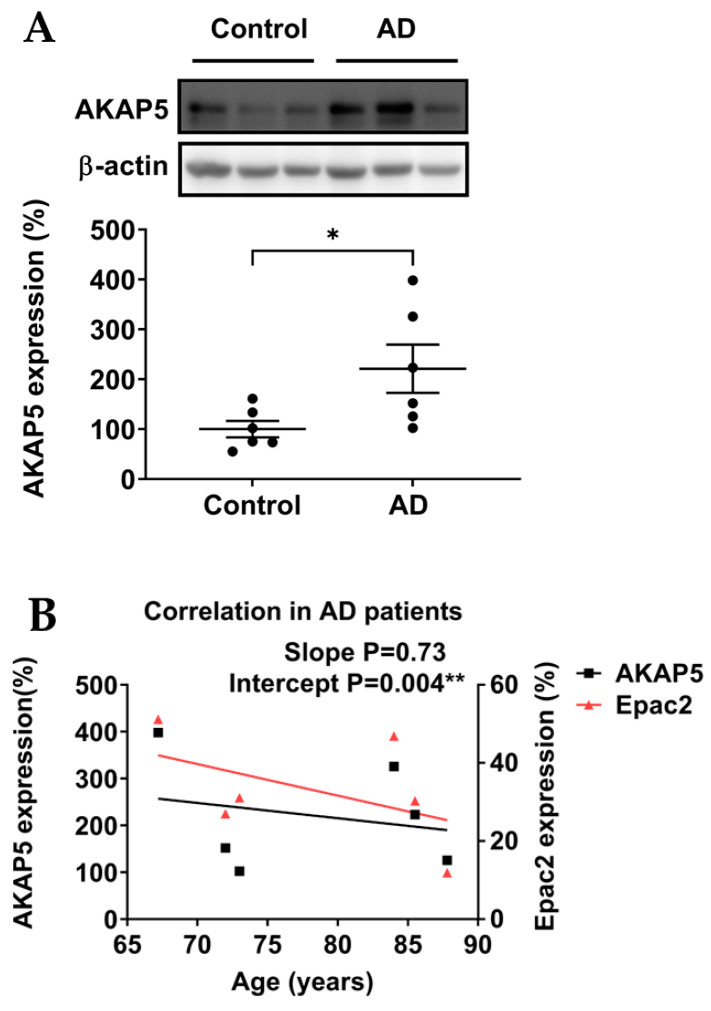
Relative protein expression levels of AKAP5. (**A**) In postmortem samples of AD patients, the protein expression of AKAP5 was significantly upregulated. (**B**) Protein expression of AKAP5 and Epac2 gradually decreased with age in postmortem samples of AD patients. (**C**) In the experimental AD model *J20 mice*, the expression profile of AKAP5 was not significantly altered. Quantification and representative blots are shown. Healthy controls n = 6; AD patients n = 5–6; *WT mice* = 10; *J20 mice* = 10. Data are expressed as mean ± SEM. Unpaired student’s T test and linear regression were used to study protein expressions and the trends of protein expressions with age, respectively (* *p* ≤ 0.05, ** *p* ≤ 0.01).

**Figure 4 biomedicines-11-02096-f004:**
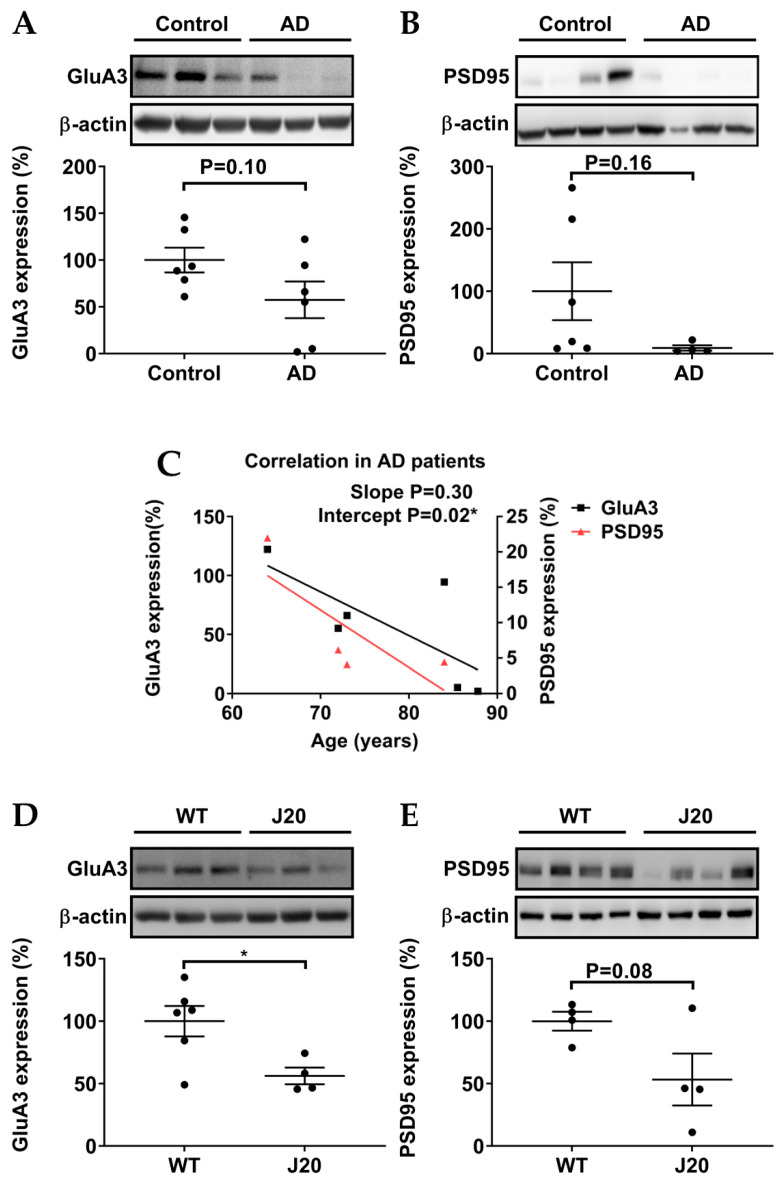
Relative expression levels of GluA3 and PSD95. (**A**,**B**) In postmortem samples of AD patients, the protein expression of GluA3 and PSD95 tended to be downregulated. (**C**) Protein expression of GluA3 and PSD95 tended to decrease with age in postmortem samples of AD patients and healthy controls. (**D**,**E**) In the experimental AD model *J20 mice*, the protein expression of GluA3 was significantly downregulated, whereas the protein expression of PSD95 tended to be downregulated. Quantification and representative blots are shown. Healthy controls n = 6; AD patients n = 4–6; *WT mice* = 4–5; *J20 mice* = 4. Data are expressed as mean ± SEM. Unpaired student’s T test and linear regression were used to study protein expressions and the trends of protein expressions with age, respectively (* *p* ≤ 0.05).

**Figure 5 biomedicines-11-02096-f005:**
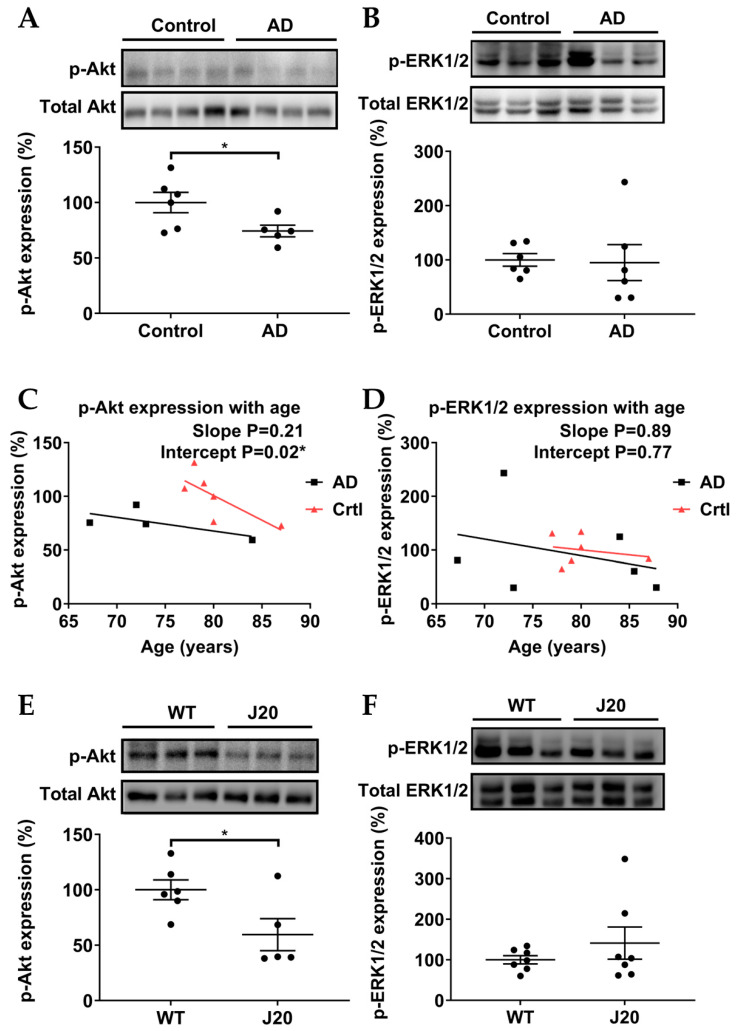
Relative expression levels of p-Akt and p-ERK. (**A**,**B**) p-Akt was significantly downregulated in postmortem samples of AD while p-ERK was not significantly altered. (**C**,**D**) p-Akt and p-ERK were downregulated with age in postmortem samples of AD patients and healthy controls (Crtl). (**E**,**F**) p-Akt was significantly downregulated in experimental AD model *J20 mice* while p-ERK was not significantly altered. Quantification and representative blots are shown. Healthy controls n = 6; AD patients n = 5–6; *WT mice* = 6–7; *J20 mice* = 5–7. Data are expressed as mean ± SEM. Unpaired student’s T test and linear regression were used to study protein expressions and the trends of protein expressions with age, respectively (* *p* ≤ 0.05).

## Data Availability

All necessary data are included in the paper.

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
