# Peer review of "The Expression of Epac2 and GluA3 in an Alzheimer’s Disease Experimental Model and Postmortem Patient Samples"

_biomedicines, 2023, doi:10.3390/biomedicines11082096_

Round 1

Reviewer 1 Report

The expression of Epac2 and GluA3 in an Alzheimer's disease experimental model and postmortem patient samples

This article, devoted to some molecular mechanisms of the pathogenesis of Alzheimer's disease, is of great interest and reveals new disorders for pharmacological correction. The introduction is well written, the specified information contains correct references, the choice of procedure is not questioned.

Questions:

1. Line 217. The number of brain samples separated by dashes (5-6; 6-7) is indicated. How many animals/samples were in each experiment? Why is there such a scatter in the choice for different experiments, for example, line 239 - J20 = 10, and in line 267 - J20 mice = 4? 4-5 animals per group is a fairly small sample.

2. Why did we choose this particular line, and not, say, 5xFAD, which have a wider presence? Why the variation in the age of mice at 4 months?

Author Response

Dear reviewer:

We thank the reviewer for his/her positive judgment of our current manuscript. Please find enclosed a point-to-point response to the points being raised.

Kind regards

Martina Schmidt

Reviewer 2 Report

Scientific Premise: There is an understanding that memory retrieval plays a bigger role in dementia compared to memory formation, atleast during early stages of the disease. The authors have therefore chosen an aggressive model of Ab aggregation – J20 – to model their early stage dementia. The premise behind exploring early stage is with an intention to identify mechanisms that are reversible at this stage thereby preventing the cascade to progressive cognitive decline. The authors focus on the Epac2 signalosome that plays a role in synaptic plasticity and memory associated with glutamatergic signaling, particularly GluA3. The selection of the J20 mouse model is based on solid scientific premise that postmortem samples of AD patients show reduced Epac2, phospho-Akt, GluA3, PSD95 while AKAP5 and is faithfully represented in their choice of the mouse model.

Major Points:

1)    A clear systematic approach to looking at the signalosome using human postmortem samples was outlined in the first figure with detailed understanding of the decrease in Epac2 expression with age in CTRL group while the levels were significantly lower in the AD group from an earlier age and did not show any further reduction at a later age, thus building a solid premise for the study. Since they have contrasted this with Epac1, there is enough corroborative evidence for the internal consistency of the data. And the authors show in the same figure that a similar relation exists in the J20 mice – minor point – in the figure and/or figure legend, it would be beneficial if the authors indicate that the experiments were performed in hippocampi.

2)    Given that the authors have mentioned that there are changes in the frontal cortex for Epac1 and Epac2 expression – could the authors provide evidence for the expression pattern seen with J20 to complete the observation?

3)    In providing the contradictory data about how AKAP5 is changed in AD patients and not in J20 mice, the authors have improved the integrity of their observation. Perhaps using a crude synaptosomal prep may have improved the signal-to-noise ratio enough in order to see the mechanism at the synaptic level over using a whole homogenate, because there are trends and this reviewer feels that the experimental results may improve with this addition.

4)    While the conclusions and discussion by the authors is very cogent and in support of the data that they have generated, the parsimonious nature needs to be addressed using a synaptosomal fraction to ensure that the cytoplasmic reserve levels are not interfering with the data.

5)    While the authors have proposed a compensatory role for AKAP5, it may be important to validate these claims using a model that expressed Tau along with Ab, like 3xTg-AD. The authors could measure the levels at 2,3,6,9 and 12 months for AKAP5 to see whether their hypothesis of AKAP5 is proved since there is a progressively accumulation with age of Tau beginning from 7 months onwards.

Commensurate.

Author Response

(The authors gave the same response as above.)
